# Propelling and perturbing appendages together facilitate strenuous ground self-righting

**Ratan Othayoth, Chen Li\***

Department of Mechanical Engineering, Johns Hopkins University, Baltimore, United States

**Abstract** Terrestrial animals must self-right when overturned on the ground, but this locomotor task is strenuous. To do so, the discoid cockroach often pushes its wings against the ground to begin a somersault which rarely succeeds. As it repeatedly attempts this, the animal probabilistically rolls to the side to self-right. During winged self-righting, the animal flails its legs vigorously. Here, we studied whether wing opening and leg flailing together facilitate strenuous ground self-righting. Adding mass to increase hind leg flailing kinetic energy increased the animal's self-righting probability. We then developed a robot with similar strenuous self-righting behavior and used it as a physical model for systematic experiments. The robot's self-righting probability increased with wing opening and leg flailing amplitudes. A potential energy landscape model revealed that, although wing opening did not generate sufficient kinetic energy to overcome the high pitch potential energy barrier to somersault, it reduced the barrier for rolling, facilitating the small kinetic energy from leg flailing to probabilistically overcome it to self-right. The model also revealed that the stereotyped body motion during self-righting emerged from physical interaction of the body and appendages with the ground. Our work demonstrated the usefulness of potential energy landscape for modeling self-righting transitions.

**\*For correspondence:**
chen.li@jhu.edu

**Competing interests:** The authors declare that no competing interests exist.

## Introduction

Ground self-righting is a critical locomotor capability that animals must have to survive (for a review, see *Li et al., 2019*). The longer an animal is flipped over and stranded, the more susceptible it is to risks like predation, starvation, desiccation (*Steyermark and Spotila, 2001*), and limited mating success (*Penn and Jane Brockmann, 1995*). Thus, it is crucial for animals to be able to self-right at a high probability because it can mean the difference between life and death. Similarly, ground self-righting is critical for the continuous operation of mobile robots (for a review, see *Li et al., 2017*).

Ground self-righting is a strenuous task. For example, to self-right, cockroaches must overcome potential energy barriers seven times greater than the mechanical energy required per stride for steady-state, medium speed running (eight body lengths s$^{-1}$) (*Kram et al., 1997*) or, exert ground reaction forces eight times greater than that during steady-state medium speed running (five body lengths s$^{-1}$) (*Full et al., 1995*). Often, animals struggle to self-right quickly and needs multiple attempts (*Brackenbury, 1990*; *Domokos and Várkonyi, 2008*; *Full et al., 1995*; *Hoffman, 1980*; *Koppányi and Kleitman, 1927*; *Li et al., 2019*; *Silvey, 1973*) to self-right due to constraints from morphology, actuation, and the terrain (*Domokos and Várkonyi, 2008*; *Faisal and Matheson, 2001*; *Golubović et al., 2017*; *Golubović et al., 2013*; *Li et al., 2019*; *Steyermark and Spotila, 2001*).

Ground self-righting has been studied in a diversity of animals, including insects (*Brackenbury, 1990*; *Delcomyn, 1987*; *Faisal and Matheson, 2001*; *Frantsevich and Mokrushov, 1980*; *Li et al., 2019*; *Sherman et al., 1977*; *Zill, 1986*), crustaceans (*Davis, 1968*; *Silvey, 1973*), mollusks

(*Hoffman, 1980*; *Weldon and Hoffman, 1979*; *Zhang et al., 2020*), and vertebrates (*Ashe, 1970*; *Bartholomew and Caswell, 1951*; *Creery and Bland, 1980*; *Domokos and Várkonyi, 2008*; *Golubović et al., 2015*; *Koppányi and Kleitman, 1927*; *Malashichev, 2016*; *Pellis et al., 1991*; *Robins et al., 1998*; *Vince, 1986*; *Winters et al., 1986*). A diversity of strategies have been described, including using appendages such as legs, wings, tail, and neck and deforming the body substantially. Often, rather than using a single type of appendages or just deforming the body without using appendages, animals use them together to propel and perturb the body to destabilize from the upside-down state (*Brackenbury, 1990*; *Davis, 1968*; *Domokos and Várkonyi, 2008*; *Faisal and Matheson, 2001*; *Hoffman, 1980*; *Li et al., 2019*). In particular, vigorous appendage flailing is a ubiquitous behavior observed across a diversity of species (*Ashe, 1970*; *Brackenbury, 1990*; *Davis, 1968*; *Delcomyn, 1987*; *Domokos and Várkonyi, 2008*; *Faisal and Matheson, 2001*; *Full et al., 1995*; *Hoffman, 1980*; *Kleitman and Koppanyi, 1926*; *Koppányi and Kleitman, 1927*; *Li et al., 2019*; *Silvey, 1973*; *Zill, 1986*). Some of these animals also use other appendages or the body to propel against the ground (*Brackenbury, 1990*; *Davis, 1968*; *Domokos and Várkonyi, 2008*; *Faisal and Matheson, 2001*; *Hoffman, 1980*; *Li et al., 2019*), and such vigorous appendage flailing appears to be a desperate, wasteful struggle.

Here, we study how propulsive and perturbing appendages together contribute to successful strenuous ground self-righting. Our model system is the discoid cockroach's strenuous ground self-righting using wings [The discoid cockroach can also self-right using a legged strategy, by pushing its legs against the ground to rotate the body without wing use (*Full et al., 1995*; *Li et al., 2019*).] (*Li et al., 2019*; *Figure 1*, *Figure 1—video 1*). The overturned animal opens and pushes its wings against the ground in an attempt to self-right, resulting in its body pitching forward (*Figure 1Ai*). Because the two opened wings and head form a triangular base of support, in which the center of mass projection falls (*Figure 1Aii*), this intermediate state is metastable. However, wing pushing rarely pitches the animal all the way over its head to self-right (the pitch mode, *Figure 1A*, blue). Thus, the animal often opens and closes its wings (hereafter referred to as an attempt [Because we focused on winged self-righting, the definition of attempt here is different from that in the previous study (*Li et al., 2019*). There, an attempt was defined as an entire process during which the animal moves its body and appendages to eventually generate a pitching and/or rolling motion, and an attempt can have multiple wing opening and closing sequences.]) multiple times, resulting in its body repeatedly pitching up and down, but it fails to self-right (*Figure 1A*, black arrows, *Figure 1—video 2*). Eventually, the animal almost always self-rights by rolling sideways over one of the wings (the roll mode; *Figure 1Aiii'*, red). Although wings are the primary propulsive appendages in this self-righting strategy, the animal also vigorously flails its legs mediolaterally, even when body pitching nearly prevents them from reaching the ground (*Figure 1B*, dashed curves). The legs occasionally scrape the ground, the abdomen occasionally flexes and twists, and the wings often deform passively under load (*Li et al., 2019*). For simplicity, we focused on the perturbing effects of the more frequent leg flailing (but see discussion of these other perturbing motions). Another curious observation is that, although the animal can in principle rotate its body in arbitrary trajectories to self-right, the observed body motion is stereotyped (*Figure 1*; *Li et al., 2019*).

A recent potential energy landscape approach to locomotor transitions (*Othayoth et al., 2021*; *Othayoth et al., 2020*) provides a modeling framework to understand how propelling and perturbing appendages together contribute to strenuous ground self-righting. A previous study modeling ground self-righting of turtles in two dimensions (the transverse plane in which the body rolls) suggested that, when trapped in a gravitational potential energy well, modest kinetic energy from perturbing appendages (legs and neck) helps overcome the small potential energy barriers (*Domokos and Várkonyi, 2008*). A recent study of cockroaches took an initial step in expanding potential energy landscape modeling of ground self-righting to three dimensions (*Li et al., 2019*). However, due to frequent camera occlusions, this study was unable to measure the complex 3D motions of appendages and only modeled the animal as a rigid body. For turtles with a rigid shell interacting with the ground, modeling self-righting with a rigid body is a good first-order approximation. However, this approximation is no longer good for modeling winged self-righting of the discoid cockroach because wing opening will change potential energy landscape.

Inspired by these insights and limitations, we hypothesized that the discoid cockroach's wing opening reduces the barriers to be sufficiently low for small kinetic energy from leg flailing to overcome. This hypothesis predicted that the greater the wing opening and leg flailing are, the more

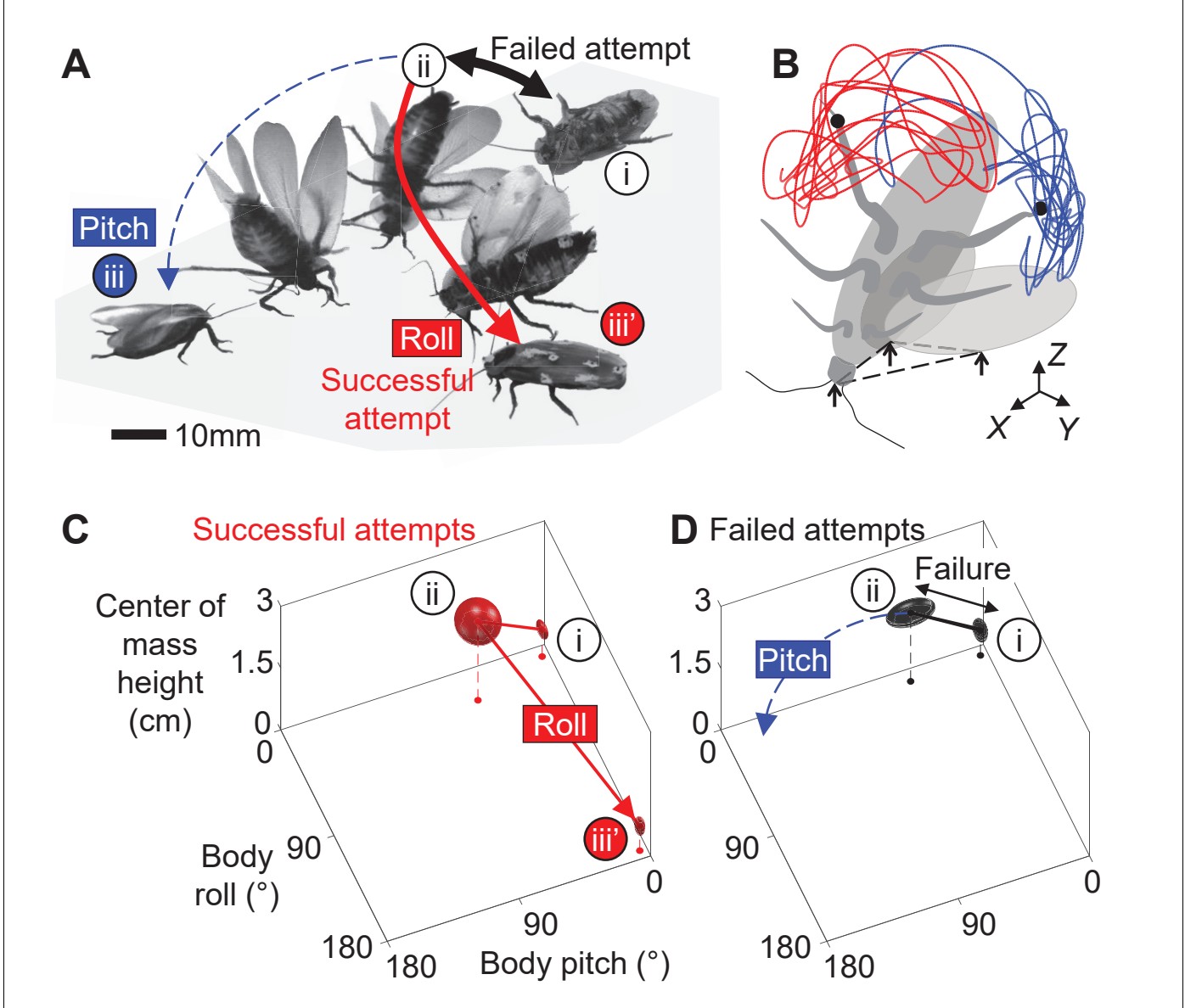

**Figure 1.** Strenuous, leg-assisted, winged ground self-righting of discoid cockroach. (A) Representative snapshots of animal successfully self-righting by pitch (blue) and roll (red) modes after multiple failed attempts (black arrow). See *Figure 1—video 1* for a typical trial, in which the animal makes multiple failed attempts to pitch over the head and eventually rolls to self-right. (B) Schematic of metastable state with a triangular base of support (dashed triangle) formed by ground contacts of head and two wing edges, with vigorous leg flailing. Red and blue curves show representative trajectories of left and right hind leg tips from a trial. *X-Y-Z* is lab frame. (C, D) Stereotyped body motion during successful (C) and failed (D) self-righting attempts in body pitch, body roll, and center of mass height space. i, ii, and iii in A, C, and D show upside-down (i), metastable (ii), and upright (iii, iii') states, respectively. Ellipsoids show means (center of ellipsoid) ± s.d. (principal semi-axis lengths of ellipsoid) of body pitch, body roll, and center of mass height at the beginning, highest center of mass height, and end of the attempt. For failed attempts, the upside-down state at the end of the attempts is not shown because it overlaps with the upside-down state at the start of the attempts (i). Data from *Li et al., 2019*.

The online version of this article includes the following video(s) for figure 1:

**Figure 1—video 1.** Strenuous leg-assisted, winged self-righting with multiple failed attempts.
https://elifesciences.org/articles/60233#fig1video1

**Figure 1—video 2.** A discoid cockroach using wing opening and leg flailing together during strenuous winged self-righting.
https://elifesciences.org/articles/60233#fig1video2

likely self-righting is to occur. We first tested this prediction in the animal, by directly modifying the hind leg inertia to increase kinetic energy from leg flailing (*Figure 2A*) and studying how it impacted self-righting probability. Then, we developed a robotic physical model (*Figure 2B*) to systematically test the prediction using repeatable experiments over a wide range of wing opening and leg oscillation amplitudes. In addition, we modeled the escape from the metastable state to self-right as a probabilistic barrier-crossing transition on an evolving potential energy landscape of the self-deforming robot/animal, facilitated by kinetic energy. The landscape is the gravitational potential energy of the robot in its body pitch-roll space. Because self-righting could in principle occur via both roll and pitch modes, we analyzed the potential energy barriers on landscape and the kinetic energy from wing opening (primary propulsion) and leg flailing (secondary perturbation) along roll and pitch directions. Considering the effects of wing opening and leg flailing separately gave new insight into the physical mechanism of self-righting. Finally, we examined whether the observed stereotypy of the animal's body motion can be explained by the potential energy landscape.

We designed and controlled our robotic physical model to achieve similar, strenuous self-righting behavior as the animal's, where both wing and leg use are crucial (see Discussion). The robot consisted of a head, two wings, a leg, and motors to actuate the wings and leg (*Figure 2B*, *Table 1*). To emulate the animal's wing opening, both robot wings opened by rolling and pitching about the body by the same angle (defined as wing opening amplitude, $\theta_{wing}$; *Figure 2B*, *Figure 2—figure supplement 1*, *Figure 2—video 1*). To simplify leg flailing of the animal, the robot used a pendulum leg which oscillated in the coronal plane by the same angle to both sides (defined as leg oscillation amplitude, $\theta_{leg}$; *Figure 2B*, *Figure 2—figure supplement 1*, *Figure 2—video 1*). We opened and closed the robot's wings (hereafter referred to as an attempt) repeatedly while oscillating its legs to generate repeated attempts observed in the animal. The robot's leg oscillation was feedforward-controlled, considering that the animal's leg flailing motion did not correlate with wing opening motion (see Materials and methods for details). Sufficiently large or sufficiently asymmetric wing opening alone guarantees self-righting (*Li et al., 2017*; *Li et al., 2016*). Here, to study the effect of using both wings and legs under the most strenuous condition, we chose to open both wings symmetrically and only used sufficiently small $\theta_{wing}$ with which the robot did not always self-right with

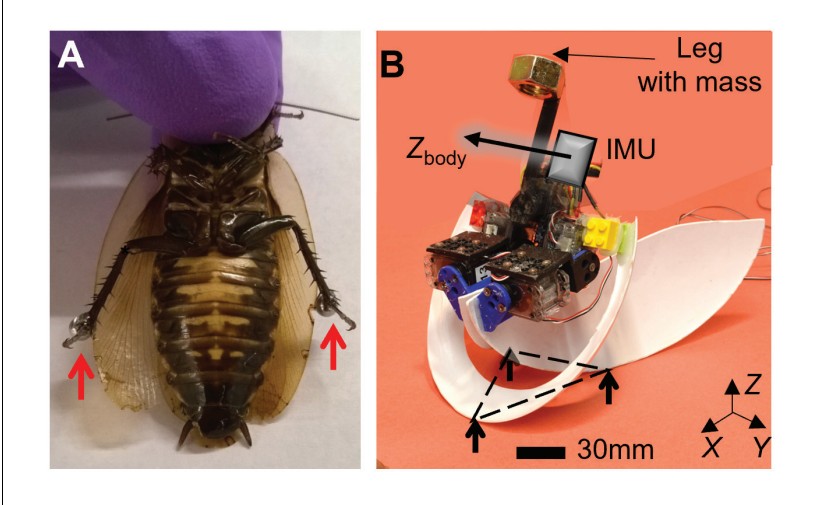

**Figure 2.** Animal leg modification and robotic physical model. (**A**) Discoid cockroach with modified hind legs with stainless steel spheres attached. (**B**) Robotic physical model in metastable state with a triangular base of support (dashed triangle), formed by ground contacts of head and two wing edges. Black arrow shows body Z-axis, $Z_{body}$. The online version of this article includes the following video and figure supplement(s) for figure 2:

**Figure supplement 1.** Robot wing and leg actuation and body orientation measurement.

**Figure 2—video 1.** Wing and leg actuation of robotic physical model.

https://elifesciences.org/articles/60233#fig2video1

**Table 1.** Mass distribution of the robot.

| Component | Mass (g) |
|---|---|
| Head | 13.4 |
| Leg rod | 4.3 |
| Leg added mass | 51.5 |
| Leg motor | 28.6 |
| Two wings | 57.4 |
| Two wing pitch motors | 56.0 |
| Two wing roll motors | 48.8 |
| Total | 260.0 |

wing opening alone. We emphasize that our goal was not to simply achieve successful self-righting in a robot.

We chose to focus potential energy landscape modeling on the robotic physical model because it offers two advantages. First, the animal's complex 3D motion with many degrees of freedom is difficult to quantify. It would take ~540 hr (~12 working weeks) to track our animal dataset (~5 s per trial at 200 frames s$^{-1}$, with three markers on the body, each wing, and each leg) to quantify 3D motion required for calculating the potential energy landscape. In addition, wing motion is often impossible to quantify due to occlusion under the body. By contrast, the robot's simpler mechanical design, controlled actuation, and an onboard inertial measurement unit (IMU) sensor allowed easier reconstruction of its 3D motion. Second, the animal's wing opening and leg flailing are highly variable (*Xuan and Li, 2020a*) and cannot be controlled. This results in the potential energy landscape varying substantially from trial to trial and makes it difficult to evaluate how the system behaved probabilistically on the landscape. By contrast, the robot's controlled variation of wing opening and leg flailing allowed us to do so. Considering that body rolling is induced by centrifugal force from leg flailing, we compared the ratio of leg centrifugal force to leg gravitational force between the animal and robot and verified they are dynamically similar (see Materials and methods for details). In addition, because the animal and robot are geometrically similar, their potential energy barriers also scale as expected (*Table 2*). Thus, the physical principles discovered for the robot are applicable to the animal.

**Table 2.** Comparison between animal and robot.

| Parameter | | Animal | Robot | Ratio |
|---|---|---|---|---|
| Body length 2$a$ (mm) | | 53 | 260 | 4.9 |
| Body width 2$b$ (mm) | | 23 | 220 | 9.6 |
| Body thickness 2$c$ (mm) | | 8 | 43 | 5.4 |
| Mass attached to leg (g) | | 0.14 | 51.5 | 368 |
| Total mass $m$* (g) | | 2.84 | 260 | 90 |
| Density $\rho$ ($\times 10^{-3}$ g mm$^{-3}$) | | 0.88 | 2.05 | 2.3 |
| Expected length scale factor $(m/\rho)^{1/3}$ | | 1.47 | 5.06 | 3.4 |
| Expected potential energy scale factor $m^{4/3}/\rho^{1/3}$ | | 4.28 | 1306 | 305 |
| Maximum pitch potential energy barrier (mJ) | | 0.58 | 282 | 486 |
| Maximum roll potential energy barrier (mJ) | | 0.19 | 244 | 1284 |
| Froude number for leg flailing $Fr$ | Intact legs | 0.37 | 0.78 | 2.1 |
| | Modified legs | 1.27 | | 0.61 |

*Includes mass attached to the legs.

## Results

### Leg flailing facilitates animal winged self-righting

As leg modification increased the animal's average kinetic energy in both pitch and roll directions (by 2 and 10 times, respectively; *Figure 3A*, *Figure 3—figure supplement 1*; p < 0.05, ANOVA), its probability of self-righting using wings increased (*Figure 3B*, *Figure 3—video 1*; p < 0.0001, mixed-effects ANOVA). These observations supported our hypothesis. Leg modification did not change the animal's relative preference of using winged and legged self-righting strategies (*Figure 3—figure supplement 2*). In addition, wing opening and leg flailing did not show temporal correlation. Furthermore, the approximate time period of leg flailing (100 ms) was comparable to combined sensory feedback (6–40 ms; *Ritzmann et al., 2012*) and neuromuscular (45 ms; *Sponberg and Full, 2008*) delays. These, combined with the fact that previous studies observed minimal proprioceptive sensory input from legs during flailing (*Camhi, 1977*; *Delcomyn, 1987*; *Zill, 1986*), indicate that leg flailing was more feedforward-driven than a feedback-controlled reflex coordinated with wing opening (*Figure 3—figure supplement 1*). Moreover, large trial-to-trial variations in the number of attempts required to self-right showed that the animal's self-righting was stochastic (*Figure 3—figure supplement 3*).

### Wing opening and leg flailing together facilitate robot self-righting

The robot's self-righting performance increased with both wing opening amplitude $\theta_{wing}$ and leg oscillation amplitude $\theta_{leg}$ (*Figure 4*). Similar to the animal, the robot's self-righting was stochastic, with large trial-to-trial variation in the number of attempts required to self-right and body pitching and rolling motions (*Figure 6*, *Figure 6—figure supplement 1*). For each $\theta_{wing}$ tested, as $\theta_{leg}$ increased, average roll kinetic energy increased (*Figure 4B*; p < 0.0001, ANOVA) and the robot's self-righting probability increased (*Figure 4C*; p < 0.0001, nominal logistic regression), reaching one at higher $\theta_{leg}$. Meanwhile, the number of attempts required for self-righting decreased (*Figure 4D*, *Figure 4—video 1*; p < 0.05, ANOVA). At the maximal $\theta_{leg}$ tested (45°), the robot always self-righted (*Figure 4C*) and always did so in the first wing opening attempt (*Figure 4D*). Together, these results

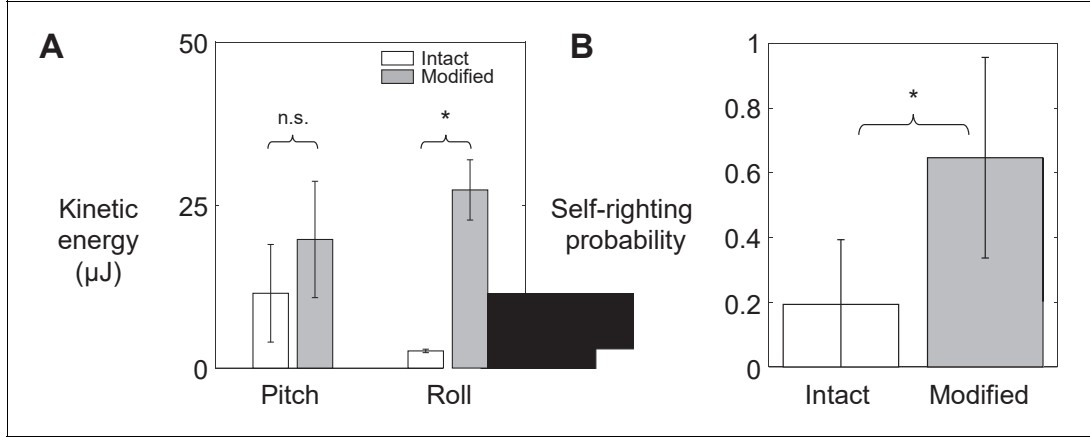

**Figure 3.** Animal's kinetic energy and self-righting probability. Comparison of (**A**) average pitch and roll kinetic energy and (**B**) self-righting probability between intact animals and animals with modified hind legs. Error bars show ± s.d. Asterisk indicates a significant difference (p < 0.05) and n.s. indicates none. Statistical tests: Pitch kinetic energy: p = 0.34, $F_{1, 1}$ = 1.53, ANOVA. Roll kinetic energy: p = 0.02, $F_{1, 1}$ = 50.35, ANOVA. Probability: p < 0.0001, $F_{1, 29}$ = 93.38, mixed-effects ANOVA. Sample size: (**A**) N = 2 animals, n = 2 trials. (**B**) Intact: N = 30 animals, n = 150 trials. Modified: N = 30 animals, n = 150 trials.

The online version of this article includes the following video and figure supplement(s) for figure 3:

**Figure supplement 1.** Animal kinetic energy calculation.

**Figure supplement 2.** Comparison of average percentage of time spent on winged and legged self-righting attempts between animals with intact and modified legs.

**Figure supplement 3.** Correlation between animal's body and leg motion.

**Figure 3—video 1.** Leg flailing kinetic energy facilitates winged self-righting of animal.

https://elifesciences.org/articles/60233#fig3video1

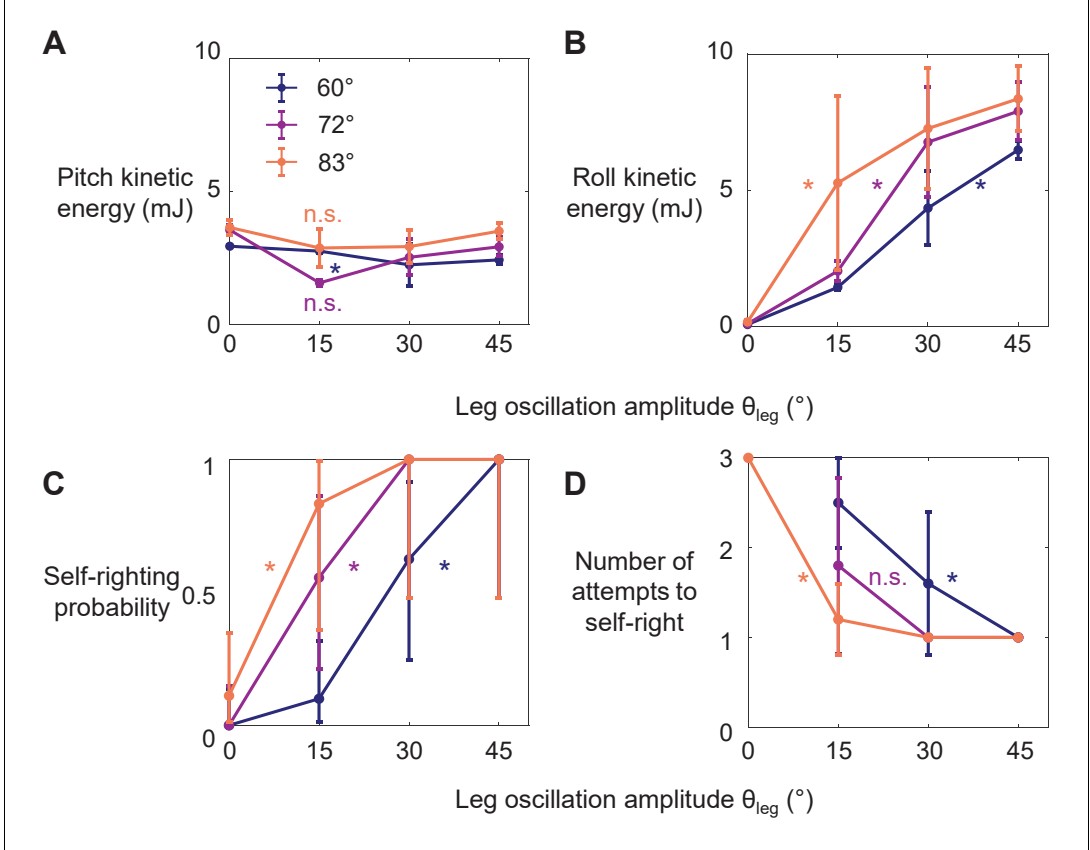

**Figure 4.** Robot's kinetic energy and self-righting performance. (A, B) Average pitch and roll kinetic energy during self-righting as a function of leg oscillation amplitude $\theta_{leg}$ at different wing opening amplitudes $\theta_{wing}$. (C, D) Self-righting probability and average number of attempts required to self-right as a function of $\theta_{leg}$ at different $\theta_{wing}$. Error bars in A, B, and D are ± s.d., and those in C are confidence intervals of 95%. Asterisks indicate a significant dependence ($p < 0.05$) on $\theta_{leg}$ at a given $\theta_{wing}$ and n.s. indicates none. See *Figure 4—source data 1* for details of statistical tests. Sample size: Kinetic energy: $n = 20$ attempts at each wing opening amplitude. Self-righting probability and number of attempts: $n = 58$, 42, and 34 attempts at $\theta_{wing} = 60°$, 72°, and 83°. For kinetic energy, only the first attempt from each trial is used to measure the average to avoid bias from large pitching or rolling motion during subsequent attempts that self-right.

The online version of this article includes the following video and source data for figure 4:

**Source data 1.** Statistical test results for *Figure 4A–D*.

**Figure 4—video 1.** Wing opening and leg flailing together facilitate winged self-righting of robot.

https://elifesciences.org/articles/60233#fig4video1

---

demonstrated that wing opening and leg flailing together facilitate the robot's self-righting performance over the wide range of parameter space tested.

## Robot self-righting resembles animal's

The robot's winged self-righting behavior resembled that of the discoid cockroach in multiple aspects (*Figures 1*, *5A*, *Figure 5—figure supplement 1*). First, it often took the robot multiple attempts (*Figure 4D*) to self-right probabilistically (*Figure 4C*). In addition, as the wings opened, the robot's body pitched up (*Figure 5Ai*), and the head and two opened wings formed a triangular base of support in which the center of mass projection fell (metastable state, *Figure 5Aii*). In failed attempts, after the wings opened fully, the robot was unable to escape this metastable state by either pitching over the head or rolling sideways and fell back to the ground upside-down as the wings closed (*Figure 5A*). In successful attempts, the robot escaped the metastable state and always self-righted by rolling to either side (*Figure 5Aiii'*, red). Moreover, the robot never lifted off the ground during self-righting. Finally, the robot's motion trajectories in the space of body pitch, roll, and center of mass height were stereotyped for both failed and successful attempts (*Figure 6*,

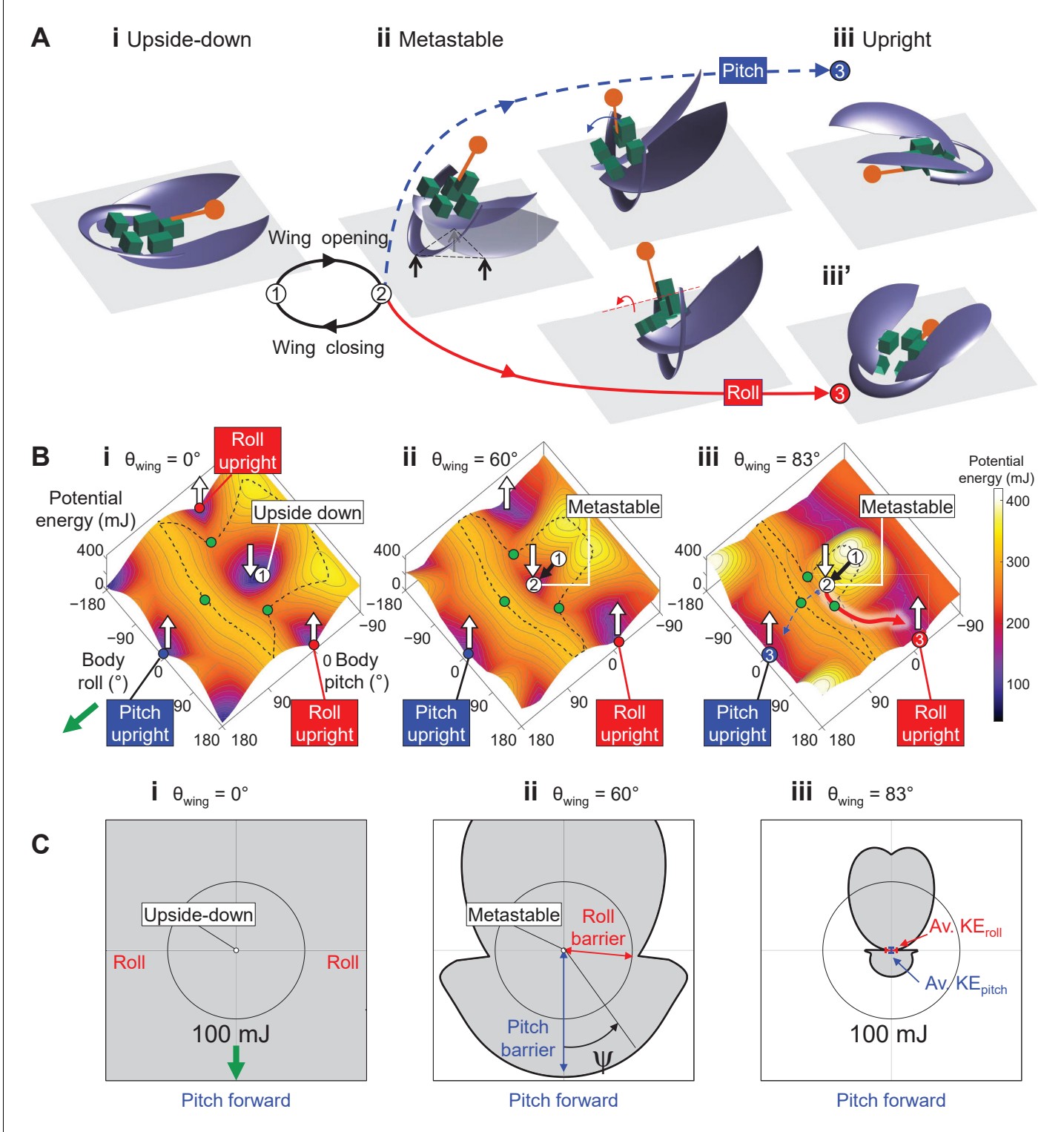

**Figure 5.** Robot's self-righting motion and potential energy landscape. (**A**) Snapshots of reconstructed robot upside-down (i), in metastable state (ii), self-righting by pitch (iii) and roll (iii') modes, and upright afterward (iv, iv'). (**B**) Snapshots of potential energy landscape at different wing opening angles corresponding to (**A**) i, ii, iii. Dashed curves are boundary of upside-down/metastable basin. Green dots show saddles between metastable basin and the three upright basins. Gray curves show constant potential energy contours. Black, dashed blue, and red curves are representative trajectories of being attracted to and trapped in metastable basin, self-righting by pitch mode, and self-righting by roll mode, respectively. i, ii, iii in (**A**, **B**) show upside-down (1), metastable (2), and upright (3iii, iii') states, respectively. (**C**) Polar plot of potential energy barrier to escape from upside-down or

*Figure 5 continued on next page*

*Figure 5 continued*

metastable local minimum along all directions in pitch-roll space. $\psi$ is polar angle defining direction of escape in body pitch-roll space. Green arrow in (i) shows direction of upright minima at pitch = 180° ($\psi$ = 0°). Black circle shows scale of energy barrier (100 mJ). Blue and red arrows in (ii) define pitch and roll potential energy barriers. Blue and red error bars in (iii) show average maximal pitch and roll kinetic energy, respectively.

The online version of this article includes the following video and figure supplement(s) for figure 5:

**Figure supplement 1.** Animal's potential energy landscape.
**Figure 5—video 1.** Robot potential energy landscape modeling.
https://elifesciences.org/articles/60233#fig5video1
**Figure 5—video 2.** Robot state trajectory on potential energy landscape.
https://elifesciences.org/articles/60233#fig5video2
**Figure 5—video 3.** Bifurcation diagram for animal's potential energy landscape.
https://elifesciences.org/articles/60233#fig5video3

*Figure 6—figure supplement 1*), although they are also stochastic with trial-to-trial variations in body pitch and roll.

## Robot and animal have similar evolving potential energy landscapes

For both the animal and robot, the potential energy landscape over body pitch-roll space was similar in shape, and both changed in a similar fashion as the wings opened (*Figure 5*, *Figure 5—figure supplement 1*). This is expected because the animal and robot were geometrically similar (*Table 2*). When the wings were fully closed, the potential energy landscape had a local minimum at near zero body pitch and roll (*Figure 5Bi*, *Figure 5—video 1*, white dot, *Figure 5—video 1*, top right). This is because either pitching or rolling of the body from being upside-down increases center of mass height and thus gravitational potential energy. Hereafter, we refer to this local minimum basin as the upside-down basin. The landscape also had three other local minima corresponding to the body being upright. [There was a fourth upright basin that can be reached by pitching downward to somersault backward, centered around a body pitch within [−180°, −162°] as the wing opening angle changed. However, such self-righting motion was not observed in the animal or robot.] One local minimum at (body pitch, roll) = (180°, 0°) could be reached from the upside-down basin by pitching forward (*Figure 5A*, blue dot). Two local minima at (body pitch, roll) = (0°, ±180°) could be reached by rolling left or right (*Figure 5Aiii'*, *Figure 5—videos 1* and *2*, red and blue curves are for roll and pitch modes, respectively). Hereafter, we refer to these basins as pitch and roll upright basins, respectively. [When the wings are fully closed, the potential energy of all three upright basins were 1.5× that of the upside-down basin.] Transition from one basin to another required overcoming the potential energy barrier separating them (*Figure 5B*, dashed black curve). As the wings opened, both the robot's and animal's potential energy landscape and its equilibria changed (*Figure 5B*, *Figure 5—figure supplement 1*). The upside-down basin evolved [The system's potential energy landscape is high-dimensional. Here, we considered potential energy as a function of body pitch, body roll, and wing opening. When plotted over the body pitch-roll space, the landscape appears to evolve as wing opening changed.] into a metastable basin around a local minimum with a positive pitch and zero roll (*Figure 5Bii*, *Figure 5—figure supplement 1Aii*, white dot). This local minimum corresponded to the metastable state with the triangular base of support (*Figures 1B*, *5Aii*). The more the wings opened, the higher the pitch of this local minimum was. To self-right via either the pitch (*Figure 5Aiii*, *Figure 1Aiii*) or roll (*Figure 5Aiii'*, *Figure 1Aiii'*) mode, the system state must escape from the metastable basin to reach either the pitch or a roll upright basin (e.g., *Figure 5Biii*, blue and red curves).

## Self-righting transitions are destabilizing, barrier-crossing transitions on landscape

Reconstruction of the robot's 3D motion on the potential energy landscape revealed that its self-righting transitions are probabilistic barrier-crossing transitions (*Figure 6*, *Figure 6—video 1*). Except when the robot was upright, upside-down, or metastable, it was always statically unstable and its system state was strongly attracted to one of these three local minima basins. At the beginning of each attempt, the system state was in the upside-down basin. As the wings opened, it was

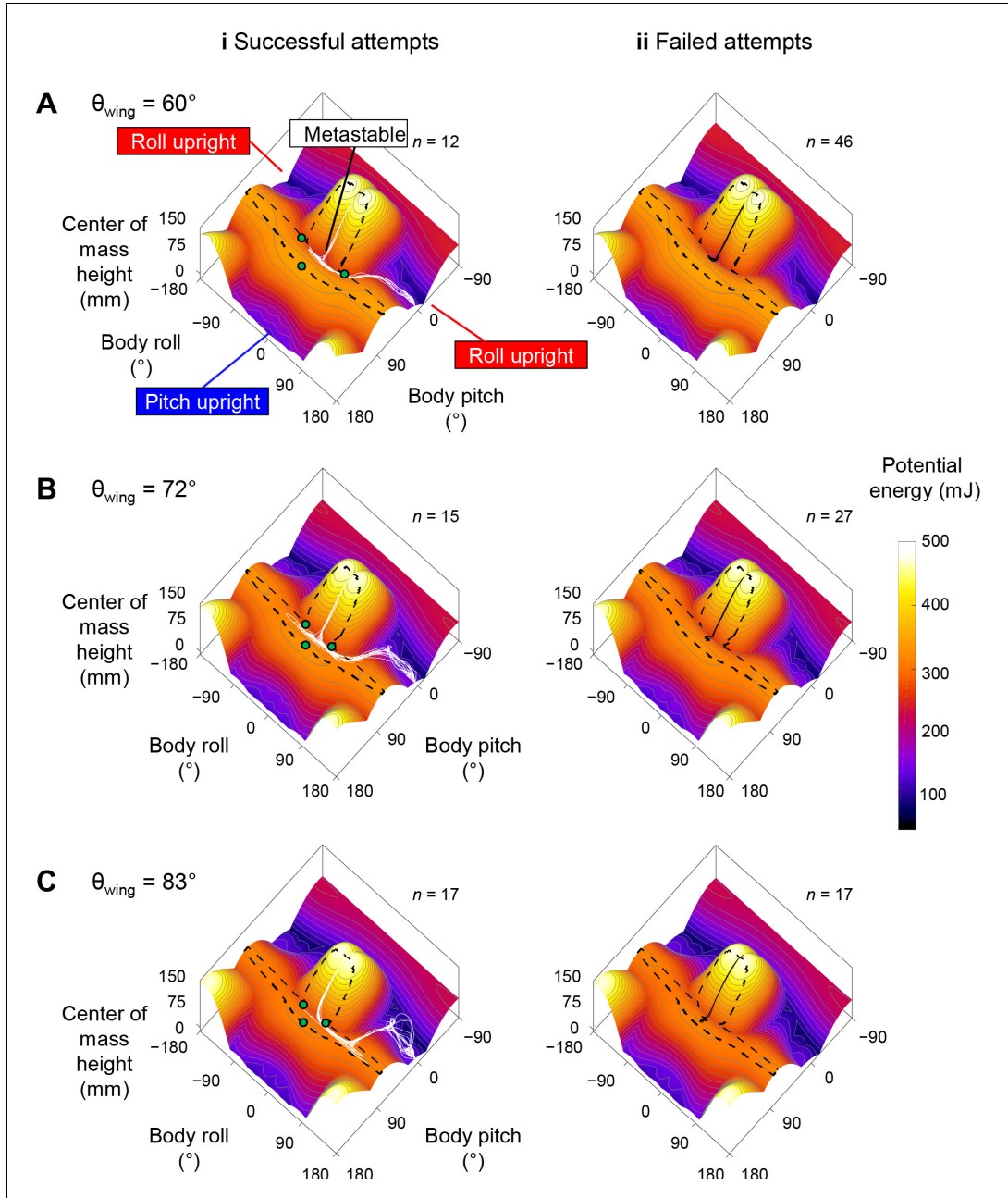

**Figure 6.** Robot state trajectories on potential energy landscape. (**A**) $\theta_{wing}$ = 60°. (**B**) $\theta_{wing}$ = 72°. (**C**) $\theta_{wing}$ = 83°. Columns i and ii show successful (white) and failed (black) self-righting attempts, respectively. *n* is the number of successful or failed attempts at each $\theta_{wing}$. Note that only the end point of the trajectory, which represented the current state, showed the actual potential energy of the system at the corresponding wing opening angle. The rest of the visualized trajectory showed how body pitch and roll evolved but, for visualization purpose, was simply projected on the landscape surface. Gray lines show energy contours. Green dots show saddles between metastable basin and the three upright basins.

The online version of this article includes the following video and figure supplement(s) for figure 6:

**Figure supplement 1.** Robot's stereotyped body motion during self-righting.

**Figure 6—video 1.** Robot state trajectory ensemble on potential energy landscape.

https://elifesciences.org/articles/60233#fig6video1

attracted toward the metastable basin that emerged. In failed attempts, the system state was trapped in the metastable basin and unable to escape it (**Figure 6**, black curves). In successful attempts, it crossed a potential energy barrier (**Figure 5B**, dashed black curve) to escape the

metastable basin and reach a roll upright basin (*Figure 6*, *Figure 6—video 1*, white curves). These observations are in accord with the animal's center of mass height measurements at the beginning, maximal pitch, and end of each attempt from the previous study (*Li et al., 2019*) projected onto the animal's potential energy landscape (*Figure 2C,D*).

## Self-righting via rolling overcomes smaller barrier than via pitching

For both the animal and robot, the potential energy landscape model allowed us to quantify the potential energy barrier for self-righting via the pitch and roll modes. The barrier to escape the metastable state to self-right varied with the direction along which the system moved in the body pitch-roll space (*Figures 5C*, *7C*, *Figure 7—figure supplement 1*). We defined the pitch and roll barriers as the minimal barriers to escape from the metastable local minimum toward the pitch and roll upright basins (*Figure 5C*, blue and red arrow). At all wing opening angles up to 90°, the roll barrier was always lower than the pitch barrier (*Figures 5C*, *7C*, *Figure 7—figure supplement 1C*).

## Barrier reduction by wing opening facilitates self-righting via rolling

For both the animal and robot, as wing opening angle increased, both the pitch and roll barrier decreased monotonically (*Figure 7C*, *Figure 5—figure supplement 1*, bottom left). As the wings opened to the range of $\theta_{wing}$ tested (*Figure 7C*, gray band), the pitch barrier was still much greater than the average pitch kinetic energy (*Figure 7C*, *Figure 7—figure supplement 1C*, solid curve vs. dashed blue line). By contrast, the roll barrier was lowered to a similar level as the average roll kinetic energy (*Figure 7C*, solid curve vs. dashed red line). This explained why the modified animal, with its higher average kinetic energy, self-righted at a higher probability than the intact animal (*Figure 7—figure supplement 1* solid vs. dashed lines). These findings demonstrated that, even though wing opening did not generate sufficient kinetic energy to self-right by pitching (*Figure 7C*), it reduced the roll barrier so that self-righting became possible using small, perturbing roll kinetic energy from leg flailing.

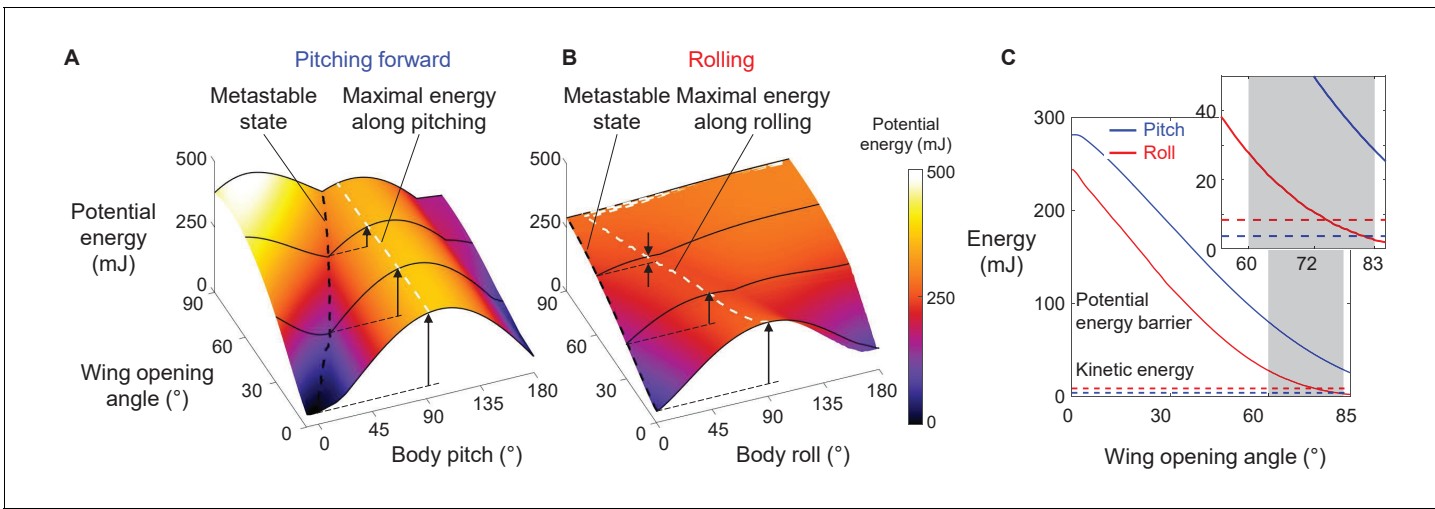

**Figure 7.** Robot's potential energy barriers for self-righting via pitch and roll modes. (**A**) Potential energy during self-righting via pitch mode as a function of body pitch and wing opening angle. (**B**) Potential energy during self-righting via roll mode as a function of body roll and wing opening angle. Dashed black curves in A and B show energy of metastable state. Dashed white curves in A and B shows maximal energy when pitching forward or rolling from metastable state, respectively. Vertical upward arrows define pitch (**A**) and roll (**B**) barriers at a few representative wing opening angles. (**C**) Pitch (blue) and roll (red) barrier as a function of wing opening angle. Blue and red dashed lines show average maximal pitch and roll kinetic energy, respectively. Gray band shows range of wing opening amplitudes tested. Inset shows the same data magnified to better show kinetic energy.

The online version of this article includes the following source data and figure supplement(s) for figure 7:

**Source data 1.** Statistical test results for *Figure 7—figure supplements 2* and *3*.

**Figure supplement 1.** Animal's potential energy barriers for self-righting via pitch and roll modes.

**Figure supplement 2.** Comparison between robot's roll kinetic energy and roll potential energy barrier.

**Figure supplement 3.** Comparison between robot's pitch kinetic energy and pitch potential energy barrier.

To further confirm this, we compared the robot's kinetic energy with potential energy barrier along the pitch and roll directions respectively during each attempt (*Figure 7—figure supplements 2* and *3*). The robot's pitch kinetic energy was insufficient to overcome even the reduced pitch barrier in both failed and successful attempts (*Figure 7—figure supplement 3*). By contrast, as wing opening and leg flailing amplitudes increased, the robot's roll kinetic energy more substantially exceeded the roll barrier during successful attempts (*Figure 7—figure supplement 2*; p < 0.001, nominal logistic regression), and the surplus enabled it to self-right via rolling.

## Discussion

We integrated animal experiments, robotic physical modeling, and potential energy landscape modeling to discover the physical principles of how the discoid cockroach uses propelling and perturbing appendages (wings and legs, respectively) together to achieve strenuous ground self-righting. Ground self-righting transitions are stochastic, destabilizing barrier-crossing transitions on a potential energy landscape. Even though propelling appendages cannot generate sufficient kinetic energy to cross the high potential energy barrier of this strenuous locomotor task, they modify the landscape and lower the barriers in other directions sufficiently so that kinetic energy from perturbing appendages can help cross them probabilistically to self-right. Compared to only using propelling or perturbing appendages alone, using them together makes self-righting more probable and reduces the number of attempts required, increasing the chance of survival.

Although the intact animal's average kinetic energy from hind leg flailing was not sufficient to overcome the potential barrier at the range of wing opening observed, it still self-righted at a small but finite probability (*Figure 3B*). This was likely because of the additional kinetic energy from flailing of fore and mid legs, small forces from legs scraping the ground, as well as abdominal flexion and twisting and passive wing deformation under load (*Li et al., 2019*), both of which induce lateral asymmetry and tilts the potential energy landscape toward one side and lowers the roll barrier. This consideration further demonstrates the usefulness of co-opting a variety of appendages for propulsion and perturbation simultaneously to achieve strenuous ground self-righting. Such exaptation (*Gould and Vrba, 1982*) of multiple types of appendages that evolved primarily for other locomotor functions for self-righting is likely a general behavioral adaptation and should be adopted by terrestrial robots.

### Stereotyped motion emerges from physical interaction constraint

Our landscape modeling demonstrated that the stereotyped body motion during strenuous leg-assisted, winged self-righting in both the animal and robot is strongly constrained by physical interaction of the body and appendages with the environment. The stereotyped repeated body pitching up and down during failed attempts and rolling during successful attempts directly results from the strong attraction of the system state to the landscape basins, which directly arise from physical interaction of body/appendages with the ground. This finding suggested that potential energy landscape modeling can be used to understand stereotyped ground self-righting strategies of other species (*Ashe, 1970*; *Domokos and Várkonyi, 2008*; *Golubović et al., 2013*; *Li et al., 2019*; *O'Donnel, 2018*) and even infer those of extinct species (analogous to *Gatesy et al., 2009*). Similarly, it will inform the design and control of self-righting robots (e.g., *Caporale et al., 2020*; *Kessens et al., 2012*).

Although only demonstrated in a model system, the potential energy landscape approach can in principle be applied to more complex and different self-righting behaviors, as well as on ground of different properties (*Sasaki and Nonaka, 2016*), to understand how propelling and perturbing effects work together. For example, as the ground becomes more rugged with larger asperities, the landscape becomes more rugged with more attractive basins (*Figure 8*, *Figure 8—video 1*). In addition, for leg-assisted, winged self-righting, we can add degrees of freedom for fore and mid leg flailing, abdomen flexion and twisting, and even passive wing deformation due to load (*Li et al., 2019*) to create fine-grained potential energy landscapes to understand how these motions may emerge from physical interaction constraints. We can also understand legged self-righting by modeling how the legs and deformable abdomen (*Li et al., 2019*) affect the potential energy landscape when wings are not used. This broad applicability will be useful for comparative studies across species, strategies, and even environments, such as understanding why some cockroach species' self-righting

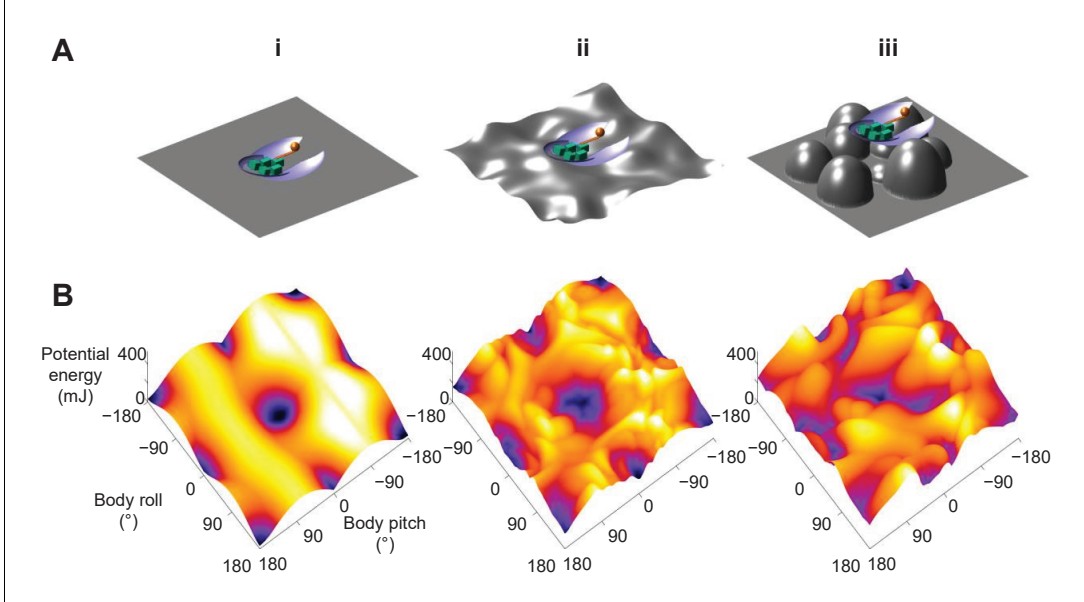

**Figure 8.** Dependence of potential energy landscape on ground geometry. (**A**) Grounds of different geometry. (i) Flat ground. (ii, iii) Uneven ground with small (ii) and large (iii) asperities compared to animal/robot size. (**B**) Potential energy landscapes for self-righting on corresponding ground. In ii and iii, landscape is not invariant to robot body translation as in i. Landscape is shown for robot at the geometric center of the terrain with wings closed. Robot shown for scale.

The online version of this article includes the following video for figure 8:

**Figure 8—video 1.** Potential energy landscape changes with ground geometry.

https://elifesciences.org/articles/60233#fig8video1

is more dynamic than others (*Li et al., 2019*). However, this approach does not apply to highly dynamic self-righting strategies, such as those using jumping (*Bolmin et al., 2017*; *Kovac et al., 2008*) where kinetic energy far exceeds the potential energy barrier.

## Toward potential energy landscape theory of self-righting transitions

The potential energy landscape model here does not describe self-righting dynamics. Recent dynamic modeling using multi-body dynamics simulations (*Xuan and Li, 2020a*) and dynamical templates (*Xuan and Li, 2020b*) in our lab revealed that wing-leg coordination affects self-righting by changing the mechanical energy budget (*Xuan and Li, 2020b*) and that the randomness in the animal's motion helps it self-right (*Xuan and Li, 2020a*). However, these approaches have their limitations: multi-body dynamic simulations are effectively experiments on a computer; dynamical templates are increasingly challenging to develop as system degrees of freedom increases. Further development of a potential energy landscape theory that adds stochastic, non-conservative forces to predict how the system 'diffuses' across landscape barriers (analogous to *Socci et al., 1996*) may be a relatively simple yet intuitive way to model probabilistic barrier-crossing dynamics.

## Materials and methods

### Animal experiments
#### Animals

We used 30 adult male *Blaberus discoidalis* cockroaches (*Figure 2A*) (Pinellas County Reptiles, St. Petersburg, FL), as females were often gravid and under different load-bearing conditions. Prior to experiments, we kept the animals in individual plastic containers at room temperature (24°C) on a 12 hr:12 hr light: dark cycle and provided water and food (rabbit pellets) ad libitum. Animals weighed $2.6 \pm 0.2$ g and measured $5.3 \pm 0.2$ cm in length, $2.3 \pm 0.1$ cm in width, and $0.8 \pm 0.1$ cm in thickness. All data are reported as mean ± s.d. unless otherwise specified.

## Leg modification

To study the effect of leg flailing, we directly modified both hind legs of the animal. We attached stainless steel spheres of diameter 0.32 cm and mass 0.14 g (5% of body weight, 180% of leg weight; *Kram et al., 1997*) (McMaster-Carr, Elmhurst, IL) to the tibia-tarsus joint of both hind legs (*Figure 2A*, *Figure 3—video 1*, right) using ultraviolet curing glue (BONDIC, Ontario, Canada). We verified that the added mass increased the average kinetic energy during leg flailing (*Figure 3—figure supplement 1*, see section 'Kinetic energy measurement').

## Experiment protocol

We used a flat, wooden surface (60 cm × 60 cm) covered with cardstock and walled with transparent acrylic sheets as the righting arena. Four 500 W work lights (Coleman Cable, Waukegan, IL) illuminated the arena for high-speed imaging. We maintained the arena at an ambient temperature of 40 ± 2℃ during experiment. We used two synchronized cameras (Fastec IL5, Fastec Imaging, San Diego, CA) at 200 frames s$^{-1}$ and 200 μs shutter time to record the self-righting maneuver from top (1200 × 1080 pixels) and side (1200 × 400 pixels) views, with a small lens aperture to maximize the focal depth of field.

For each trial, we first started video recording, held the animal upside-down by its pronotum, and gently released it from a height of ≈ 1 cm above the center of the righting arena. The small drop was to ensure that the animal did not begin leg searching, a common strategy used to self-right (*Camhi, 1977*), before it was released. The animal was given 10 s to attempt to self-right during each trial. After it self-righted or 10 s elapsed, the animal was picked up, and video recording was stopped. After each trial, we returned the animal to its container and continued testing a different animal. This way, each animal was allowed to rest for ≈ 30 min before its next trial to minimize the effects of fatigue (*Camhi, 1977*).

We tested 30 animals, each with five trials with its hind legs intact and then modified, resulting in a total of 300 accepted trials (*N* = 30 animals, *n* = 150 trials for each leg treatment). We excluded trials in which the animal collided with the walls of the righting arena or moved out of both camera views.

## Self-righting performance

For each animal trial, we watched the videos to determine whether the animal self-righted. Because the animal did not always immediately begin to self-right when placed on the arena (*Camhi, 1977*; *Li et al., 2019*), we defined the beginning of the self-righting attempt as the instant when the animal began moving its body or appendages to self-right. We defined the animal to have successfully self-righted if it attained an upright orientation with all six legs on the ground within 10 s of starting its attempt. We identified the trials in which animal succeeded in self-righting using the leg-assisted, winged strategy. For each animal and each leg treatment, we defined and measured self-righting probability as the number of trials that self-righted using winged attempts divided by the total number of trials. We counted the trials that used the legged strategy as failed. We then calculated average self-righting probability for each leg treatment by averaging across all animals.

## Preference of self-righting strategies

We verified that the animal's preference of winged and legged self-righting strategies [The discoid cockroach can also self-right using a legged strategy, by pushing its legs against the ground to rotate the body without wing use (*Full et al., 1995*; *Li et al., 2019*)] did not change with leg modification. To compare the animal's preference of winged and legged self-righting strategies before and after leg modification, for each trial, we examined the videos to identify winged and legged self-righting attempts and measured the percentage of time spent on each strategy. Then, for each leg treatment and each animal, we averaged it across all the trials from that animal. For each treatment, we then averaged across each animal to calculate the average percentage of time spent on each strategy (*Figure 3—figure supplement 2*).

## Kinetic energy measurement

To measure the animal's pitch and roll kinetic energy during self-righting, in a separate experiment, we used three high-speed cameras (Photron FASTCAM Mini UX-100) to record the animal self-

righting at 2000 frames s$^{-1}$ and a resolution of 1280 $\times$ 1024 pixels, first with its hind legs intact ($N$ = 2 animals, $n$ = 2 trials) and then modified ($N$ = 2 animals, $n$ = 2 trials).

We used DeepLabCut (**Mathis et al., 2018**) to track the tip and femur-tibia joint of both hind legs, head anterior tip, abdomen posterior tip, and body midpoint (**Figure 3—figure supplement 1A,B**). We then used Direct Linear Transformation software DLTdv5 (**Hedrick, 2008**) to reconstruct 3D motion of the tracked points and used a sixth-order Butterworth filter with a cut-off frequency of 25 Hz to filter their 3D positions.

To calculate kinetic energy, we approximated the animal body as an ellipsoid cut into two parts at 38% of total length from the anterior end, connected by a hinge joint (thorax-abdomen joint, **Figure 3—figure supplement 1A**). The smaller part represented the animal's head and thorax, and the larger part represented its abdomen. We assumed uniform mass distribution for both parts. We used the geometric center of the body parts when their fore-aft axes are aligned to approximate body center of mass (**Kram et al., 1997**). For both hind legs, we approximated the coxa-femur and tibia-tarsus segment as rigid rods. One end of the rod representing coxa-femur segment was connected to the body at the midpoint of thorax-abdomen joint, and the other end connected to the rod representing tibia-tarsus segment, both via spherical joints (**Figure 3—figure supplement 1A,B**, thick black lines connected by blue dots). For modified hind legs, we approximated the stainless steel spheres at the leg tip as a point mass attached to the free end of the tibia-tarsus rod (**Figure 3—figure supplement 1A,B**).

We defined pitch and roll kinetic energy as the sum of kinetic energy from translational and rotational velocity components from all body parts that contribute to pitching and rolling motion, respectively. We obtained pitch and roll kinetic energy by summing contributions from the body ellipsoid parts and the hind leg segments. For each part, we measured its rotational velocity components about the animal's body fore-aft ($X_{body}$) and lateral ($Y_{body}$) principal axes, and we measured the translational velocity components of its center of mass along the fore-aft and lateral directions (**Figure 3—figure supplement 1B**, red vs. blue arrows). For the sphere attached to modified leg, we measured its translational velocities. Because vertical translational velocity and yaw angular velocity did not contribute to motion along the pitch or roll direction, we did not consider them.

For each of the ellipsoid parts and rigid rods, we calculated its pitch and roll kinetic energy as follows:

$$\mathrm{KE}_{\mathrm{pitch,j}} = \frac{1}{2} I_{yy,j}\omega_{y,j}^2 + \frac{1}{2} m_j v_{x,j}^2 \tag{1}$$

$$\mathrm{KE}_{\mathrm{roll,j}} = \frac{1}{2} I_{xx,j}\omega_{x,j}^2 + \frac{1}{2} m_j v_{y,j}^2 \tag{2}$$

where $I_{xx,j}$ and $I_{yy,j}$ are the moments of inertia the $j^{th}$ object measured about the animal's body fore-aft ($X_{body}$) and lateral ($Y_{body}$) principal axes, respectively, $m_j$ is the mass of $j^{th}$ object, $\omega_{x,j}$ and $\omega_{y,j}$ are the rotational velocities of the $j^{th}$ object about body fore-aft and lateral principal axes, and $v_{x,\,j}$ and $v_{y,\,j}$ are the translational velocity of the center of mass of the $j^{th}$ object along fore-aft and lateral directions, respectively (**Figure 3—figure supplement 1B**). For both hind leg segments, we used the mass reported in **Kram et al., 1997** (0.07 g for coxa-femur segments and 0.01 g for tibia-tarsus segment). To calculate the mass of the two body parts, we assumed body density to be uniform.

We calculated the pitch and roll kinetic energy of the added spherical mass as follows:

$$\mathrm{KE}_{\mathrm{pitch,sphere}} = \frac{1}{2} m_{sphere} v_{x,sphere}^2 \tag{3}$$

$$\mathrm{KE}_{\mathrm{roll,sphere}} = \frac{1}{2} m_{sphere} v_{y,sphere}^2 \tag{4}$$

where $m_{sphere}$ is the added spherical mass, and $v_{x,\,sphere}$ and $v_{y,\,sphere}$ are the translational velocity components of the sphere along fore-aft and lateral directions, respectively. We considered kinetic energy from the added spherical mass only for animal with modified legs.

We obtained the pitch and roll kinetic energy of the intact animal from **Equations (1) and (2)** respectively. For the modified animal, we added **Equations (1) and (3)** to obtain total pitch kinetic

energy and added *Equations (2) and (4)* to obtain total roll kinetic energy. For each trial, we first averaged the measured kinetic energy along pitch and roll directions over the recorded interval (2.5 s) for each trial. Then for each leg treatment, we further averaged it across all the trials of that treatment (intact: $N = 2$ animals, $n = 2$ trials; modified: $N = 2$ animals, $n = 2$ trials).

## Relationship between wing opening and leg flailing

We examined whether the animal's leg flailing during self-righting was more feedforward-driven or more toward a feedback-controlled reflex coordinated with wing opening. To do so, we measured the correlation between wing opening and leg flailing motions as well as their self-correlations (*Figure 3—figure supplement 3*). Because wing opening was difficult to measure due to occlusion of wings by the body during self-righting, we used abdomen tip height as a proxy for wing opening, considering that abdomen tip height typically increased as wings opened. For each hind leg, we used its leg tip height as a proxy of the flailing motion (*Figure 3—figure supplement 1*). To check whether the height of abdomen tip and hind leg tips were correlated to each other and to themselves, we measured the normalized cross-correlations between each pair of these variables and the normalized autocorrelation of each of them (*Figure 3—figure supplement 3*). Normalized cross-correlation $h$ between two signals $f(t)$ and $g(t)$ is defined as:

$$h(t) = \frac{\int_{-\infty}^{\infty} f^*(\tau - t) g(\tau) d\tau}{\sqrt{\int_{-\infty}^{\infty} |f(\tau)|^2 d\tau \cdot \int_{-\infty}^{\infty} |g(\tau)|^2 d\tau}} \quad (5)$$

where $t$ is the time lag between $f(t)$ and $g(t)$ and is a variable, $\tau$ is the variable of integration, and $f^*(t)$ is the complex conjugate of $f(t)$. When $f(t) = g(t)$, $h(t)$ is the normalized autocorrelation.

All normalized cross-correlations plots lacked a prominent peak whose value was close to 1, and all normalized autocorrelations plots had a prominent peak only at zero lag. This showed that abdomen tip height did not correlate with itself or with either of the two hind leg tips heights (*Figure 3—figure supplement 3A,B,F*). This meant that wing opening and leg flailing motions were not correlated to each other during self-righting. However, the normalized cross-correlation between both hind legs had recurring oscillations as the lag increased in magnitude (*Figure 3—figure supplement 3*). This suggested that leg flailing had some rhythm, despite a large temporal variation and difference between the two hind legs (*Delcomyn, 1987*; *Sherman et al., 1977*; *Zill, 1986*).

## Robotic physical modeling

### Design and actuation

The robot consisted of a head, two wings, a leg, and four motors to actuate the wings and one to actuate the leg (*Table 1*, *Figure 2B*, *Figure 2—figure supplement 1*, *Figure 2—video 1*). The head and wings were cut from two halves of a thin ellipsoidal shell thermo-formed (Formech 508FS, Middleton, WI) from 0.16-cm-thick polystyrene plastic sheet (McMaster-Carr, Elmhurst, IL). We connected different parts using joints 3D printed using PLA (Ultimaker 2+, Geldermalsen, The Netherlands) (*Figure 2B*). We used DC servo motors (Dynamixel XL-320, ROBOTIS, Lake Forest, CA) to actuate both the wings and the leg.

### Similarity to animal

To measure the robot's 3D orientation (roll, pitch, and yaw angles), we attached an IMU (BNO055, Adafruit, New York, NY) near its center of mass determined from the robot CAD model. We used the Robot Operating System (Version: melodic) (*Quigley et al., 2009*) to send actuation signals for the wing and leg motors and record IMU data. To ensure a constant voltage for repeatable experiments, we used an external 8 V voltage source (TP3005DM, TEK Power, Montclair, CA) to power the robot. We used fine flexible wires (30 AWG, 330-DFV, Vishay Sensor, Mansfield, TX) for powering robot and sending/acquiring signals and ensured that they were loose and did not interfere with robot motion.

To examine whether the robotic physical model was similar to the animal and reasonably approximated its self-righting motion, we examined how well they were geometrically similar and their leg flailing motions were dynamically similar. To evaluate geometric similarity, we compared their dimensions. For geometrically similar objects, length $l$ should scale with mass $m$ and density $\rho$ as $l \propto (m/$

$\rho)^{1/3}$ (**Alexander, 2006**). Following this, potential energy should scale as $E \propto m \cdot (m/\rho)^{1/3} \propto m^{4/3}\rho^{-1/3}$. The robot, which was 90 times as much heavy and 2.3 times as much dense as the animal with modified legs (**Table 2**), was expected to have dimensions $(90/2.3)^{1/3} = 3.4$ times those of the animal. For the animal, $m$ includes the added mass from leg modification because we used the same for calculating the potential energy landscape. Because gravitational potential energy is proportional to mass and center of mass height, the potential energy barriers should scale by a factor of $90^{4/3} \times 2.3^{-1/3} = 305$ (**Table 2**). We found that the robot's length, thickness, and pitch potential energy barriers scaled up roughly as expected (**Table 2**). The larger scaling factor for robot's width and roll potential energy barrier is due to the robot being designed wider to make self-righting via rolling more strenuous.

To evaluate dynamic similarity between the robot and animal, we calculated Froude number for their leg flailing. Here, we used the following definition of Froude number (**Biewener, 2003**):

$$Fr = \frac{\text{Inertial force from leg flailing}}{\text{Gravitational force of leg}} = \frac{mv^2/r}{mg} = \frac{v^2}{rg} \tag{6}$$

where $m$ is the mass of the animal or robot leg(s), plus the added mass attached it for the modified animal, $v$ is the leg translational velocity along the body lateral principal axis, $g$ is gravitational acceleration, and $r$ is leg length.

We found that the Froude numbers for the robot and both the intact and modified animals were similar (within a factor of 2). This dynamic similarity demonstrated that the robot provided a good physical model for studying the animal's self-righting.

## Experiment protocol

For robot experiments, we used a level, flat, rigid wooden surface (60 cm × 60 cm) covered with sandpaper as the righting arena. We used two synchronized webcams (Logitech C920, Logitech, Newark, CA) to record the experiment from top and side views at 30 frames s$^{-1}$ and a resolution of 960 × 720 pixels (**Figure 2—figure supplement 1**). Using the onboard IMU, we recorded the robot body orientation relative to the lab coordinate system (*X-Y-Z* in **Figure 2B**) at ≈ 56 Hz and synchronized them with the motor actuation timings angles (**Figure 2—figure supplement 1**, bottom right).

Before each trial, we placed the robot upside-down (**Figure 5Ai**) on the arena, with its wings closed and leg aligned with the body midline and started video recording. We then actuated the robot to repeatedly open and close its wings at 2 Hz and oscillate its legs at 2.5 Hz to self-right. Because the animal was likely to move its leg before wings at the start of self-righting (59% of intact leg trials and 81% of modified leg trials), for non-zero robot leg oscillation amplitudes, the first wing opening was started after completing one cycle of leg oscillation (0.4 s). If the robot did not self-right after five wing opening attempts (10 s), we powered down the robot, stopped video recording, and reset the robot for the next trial. We tested self-righting performance of the robot by systematically varying leg oscillation amplitude $\theta_{\text{leg}}$ (0°, 15°, 30°, 45°) and wing opening amplitude $\theta_{\text{wing}}$ (60°, 72°, 83°). We collected five trials for each combination of $\theta_{\text{wing}}$ and $\theta_{\text{leg}}$. This resulted in a total of 60 trials with 134 attempts.

To reconstruct the robot's 3D motion, in a separate experiment, we characterized how the wing and leg actuation angles changed over time during an attempt (**Figure 2—figure supplement 1**). We attached BEETag markers (**Crall et al., 2015**) to the body frame and to each link actuated by the motors and tracked their positions using two calibrated high-speed cameras (Fastec IL5, Fastec Imaging, San Diego, CA) at 500 frame s$^{-1}$ and a resolution of 1080 × 1080 pixels, as the robot actuated its wings and legs to self-right. We obtained 3D kinematics of the markers using the Direct Linear Transformation method DLTdv5 (**Hedrick, 2008**). We then measured the rotation of the link actuated by each motor about its rotation axis as a function of time during an attempt. Because the wings were controlled to roll and pitch by the same angle, we used the average measured wing actuation profile (**Figure 2—figure supplement 1B**, dashed) of all the four motors (two for wing pitching and two for wing rolling). The actual wing opening and leg oscillation angles were smaller than the commanded (solid blue and red) due to the inertia of robot body components attached to each motor.

## Self-righting performance

We defined the beginning of the righting attempt as the instant when the robot first started opening its wings and measured this instance from the commanded motor actuation profile (*Figure 2—figure supplement 1Bii*, *Figure 2—video 1*, top right). We defined the robot to have successfully self-righted if it attained an upright orientation within 10 s (five attempts). We used the IMU to measure the projection of the gravity acceleration vector $\vec{g}$ onto the body Z-axis $\vec{Z}_{body}$ as a function of time. This allowed us to determine when the robot became upright. We then counted the number of successful and failed attempts for each trial. For each trial, we defined self-righting probability as the ratio of the number of successful attempts to the total number of attempts of that trial. At each wing opening and leg oscillation amplitude, we then averaged it across all trials of that treatment to obtain its average self-righting probability. Among all the 134 attempts observed across all 60 trials, 44 attempts succeeded (12, 15, and 17 attempts at $\theta_{wing}$ = 60°, 72°, and 83°, respectively), and 90 attempts failed (46, 27, and 17 attempts at $\theta_{wing}$ = 60°, 72°, and 83°, respectively).

## Robot 3D motion reconstruction

For each robot trial, we measured the robot's 3D orientation in the lab frame using Euler angles (yaw $\alpha$, pitch $\beta$, and roll $\gamma$, Z-Y'-X' Tait-Bryan convention). We divided each trial temporally into 0.01 s intervals and used the measured motor actuation angles and body 3D orientation (*Figure 2—figure supplement 1B,C*) at each interval to reconstruct the robot's body shape and 3D orientation, respectively. Because the IMU measured only the 3D orientation of the robot, we constrained the robot's center of mass to translate only along the vertical direction (*Figure 2B*, Z-axis of lab frame) while maintaining contact with the ground (*Figure 4—video 1*). We then used the reconstructed 3D motion of the robot to obtain the translational and rotational velocity components of all robot parts.

## Kinetic energy measurements

For each robot trial, we measured pitch and roll kinetic energy for all attempts. We defined pitch and roll kinetic energy as the kinetic energy of the entire robot due to translational and rotational velocities along body fore-aft and lateral directions, respectively. Because vertical translation and yawing do not contribute to body pitching or rolling toward self-righting, we did not consider vertical velocities or rotational velocities about the vertical axis.

Considering that the five motors, leg, and mass added to the leg could be approximated as regular, symmetric shapes with uniform mass distribution (motors and leg as solid cuboids and added mass as a solid sphere), the moment of inertia at the center of mass of each part could be directly calculated. Then, we calculated the total pitch and roll kinetic energy of the motors and leg with added mass as:

$$\text{KE}_{\text{pitch}} = \sum_{j=1}^{k} \left( \frac{1}{2} I_{yy,j} \omega_{y,j}^2 + \frac{1}{2} m_j v_{x,j}^2 \right) \tag{7}$$

$$\text{KE}_{\text{roll}} = \sum_{j=1}^{k} \left( \frac{1}{2} I_{xx,j} \omega_{x,j}^2 + \frac{1}{2} m_j v_{y,j}^2 \right) \tag{8}$$

where $j$ enumerates the five motors, leg, and mass added to the leg, $I_{xx,j}$ and $I_{yy,j}$ are the moments of inertia of object $j$ about the body fore-aft and lateral principal axes (measured at the part's center of mass), $m_j$ is the mass of object $j$, and $v_{x,j}$ and $v_{y,j}$ are translational velocities of object $j$ along fore-aft and lateral directions of robot, and $\omega_x$ and $\omega_y$ are rotational velocities of object $j$ about fore-aft and lateral directions of the robot, respectively.

For both the wings and head with complex shapes, we imported their CAD model and approximated them with uniformly distributed point mass clouds and calculated the pitch and roll kinetic energy of each part as:

$$\text{KE}_{\text{pitch,cloud}} = \frac{m}{2k} \sum_{j=1}^{k} v_{x,j}^2 \tag{9}$$

$$\mathrm{KE_{roll,cloud}} = \frac{m}{2k}\sum_{j=1}^{k} v_{y,j}^2 \qquad (10)$$

where $m$ is the total mass of the wing or head, $k$ is the number of point masses in the point cloud, and $v_{x,i}$ and $v_{y,i}$ are the velocity components of the $i$th point mass along the body fore-aft and lateral principal axes.

To obtain total pitch and roll kinetic energy, we summed the pitch and roll kinetic energy of all the parts. To compare pitch and roll kinetic energy at each combination of wing opening and leg oscillation amplitudes, we first averaged the total pitch and roll kinetic energy respectively over the phase when wings were fully open in the first attempt of each trial to avoid bias from the large rolling kinetic energy during successful self-righting in later attempts. We then averaged these temporal averages across the five trials at each combination of wing opening and leg oscillation amplitudes (*Figure 4A,B*).

## Potential energy landscape modeling
### Model definition
The gravitational potential energy of the animal or robot is:

$$E = mgz_{CoM} \qquad (11)$$

where $m$ is the total mass of the animal or robot, $g$ is gravitational acceleration, $z_{CoM}$ is center of mass height from the ground. To determine the robot's center of mass, we used a CAD model of the robot (*Figure 2A*, *Figure 2—figure supplement 1*) and measured the 3D positions and orientations of all robot body parts for a given body orientation and wing opening (see consideration of leg oscillation below). We approximated the animal body as a rigid ellipsoid, with the animal's center of mass at the body geometric center, and its wings as slices of an ellipsoidal shell. Because the animal or robot did not lift off during self-righting, in the model we constrained the lowest point of the animal or robot to be always in contact with the ground.

The potential energy depended on body pitch and roll, wing opening angle, and leg oscillation angle. Because the effect of leg oscillation was modeled as a part of kinetic energy, for simplicity, we set the leg to be held fixed in the middle when calculating the potential energy landscape. We verified that potential energy landscape did not change considerably (roll barrier changed only up to 13%) when the leg moved. Because we used Euler angles for 3D rotations, change in body yaw did not affect center of mass height. Because the robot's initial wing opening was negative ($-6°$) due to body weight, in our model calculations, we varied wing opening angle within the range [$-10°$, 90°] with a 0.5° increment. For each wing opening angle, we then varied both body pitch and roll within the range [Note that for this range, the Euler angle description of body orientations has inherent redundancies. For example, body (pitch, roll) = (0°, $-180°$) and (pitch, roll) = (0°, 180°) describe the same physical orientation of the robot. However, this does not affect our modeling and conclusions, because the system state only reaches a redundant state toward the end of self-righting when it is near-upright.] [$-180°$, 180°] with a 1° increment and calculated $z_{CoM}$ to obtain the system potential energy (*Figure 2—figure supplement 1*). Because the animal or robot did not pitch backward significantly, in the figures we do not show landscape for body pitch $<-90°$; the full landscape may be visualized using data and code provided (*Othayoth and Li, 2021a* copy archived at swh:1:rev:4454aa107b5b67428e0ae3610f06a49b053d691f *Othayoth and Li, 2021b*).

### System state trajectories on potential energy landscape
To visualize how the robot's measured system state behaved on the landscape, we first discretized each righting attempt into time intervals of 0.01 s. For each interval, we used the measured wing opening angle (*Figure 2—figure supplement 1*, dashed blue curves) to calculate the potential energy landscape (*Figure 5—video 1*, top). We then projected the measured body pitch and roll onto the landscape to obtain the system state trajectory over time (*Figure 6*, *Figure 6—video 1*). Note that only the end point of the trajectory, which represented the current state, showed the actual potential energy of the system at the corresponding wing opening angle. The rest of the

visualized trajectory showed how body pitch and roll evolved but, for visualization purpose, was simply projected on the landscape surface. The exact system state trajectories are shown in *Figure 6*.

## Potential energy barrier measurements

We measured the potential energy barrier that must be overcome to escape from metastable basin to transition to an upright basin (*Figures 5C,7*, *Figure 5—video 1*, bottom). For each wing opening angle (*Figure 7B*, dashed blue), at each time interval, we considered imaginary straight paths away from the metastable local minimum (*Figure 5B*, white dot) in the body pitch-roll space, parameterized by the polar angle $\psi$ from the positive pitch direction (body pitching up, *Figure 5Ci*). Along each path, we obtained a cross section of the landscape. Then, we defined and measured the potential energy barrier along this path as the maximal increase in potential energy in this cross section. Finally, we plotted the potential energy barrier as a function of $\psi$ (*Figure 5C*). We defined the roll barrier as the lowest potential energy barrier within $\psi = \pm$ [45˚, 135˚], because both roll upright minima always lay in this angular range. We defined the pitch barrier as the potential energy barrier at $\psi$ = 0˚ toward the pitch local minimum. Finally, we measured both pitch and roll barriers as a function of wing opening angle (*Figure 7*, *Figure 7—figure supplement 1*).

## Comparison of kinetic energy and potential energy barriers

To understand how wing opening and leg oscillation together contribute to the robot's self-righting, we compared the measured kinetic energy and potential energy barriers along both pitch and roll directions throughout each attempt. For each attempt, we measured kinetic energy minus potential energy barrier over time along both pitch and roll directions (*Figure 7—figure supplements 2* and *3A-C*). We then examined whether there was a surplus or deficit of kinetic energy to overcome the potential energy barrier in both pitch and roll directions, comparing between successful and failed attempts (*Figure 7—figure supplements 2D* and *3D*). To examine how maximal surplus varied with wing opening and leg oscillation amplitudes, for each combination of the two, we recorded the maximal surplus when the wings are held fully open in each attempt and averaged it across all attempts (*Figure 7—figure supplement 2E vs.3E* ).

## Data analysis and statistics

We tested whether the animal's percentage of time spent on winged and legged self-righting attempts and self-righting probability changed with leg modification using a mixed-effects ANOVA, with leg treatment as the fixed factor and individual as a random factor to account for individual variability. We tested whether the animal's pitch and roll kinetic energy depended on leg modification using ANOVA with leg treatment a fixed factor. We tested whether the animal's self-righting probability depended on leg treatment using a mixed-effects ANOVA with leg treatment as a fixed factor and individual as a random factor.

We tested whether the robot's self-righting probability, number of attempts required to self-right, pitch and roll kinetic energy depended on leg oscillation amplitude at each wing opening amplitude using a chi-squared test for probability and an ANOVA for the rest, with wing opening magnitude as a fixed factor. We tested whether kinetic energy minus potential energy barrier along the pitch and roll directions depended on leg oscillation amplitude at each wing opening amplitude, using ANOVAs with leg oscillation amplitude as the fixed factor. We also tested whether they depended on wing opening amplitude at each leg oscillation amplitude, using ANOVAs with wing opening amplitude as the fixed factor. To test whether kinetic energy minus potential energy barrier differed between successful and failed attempts, we used an ANOVA with the attempt outcome (success or failure) as the fixed factor. Details of statistical test results are provided in figure captions or figure supplements. All statistical tests were performed using JMP Pro 14 (SAS Institute Inc, Cary, NC).

## Acknowledgements

We thank Qihan Xuan for preliminary modeling and discussions, Sean Gart, Thomas Mitchel, and Noah Cowan for discussion and Sean Gart for help with high-speed imaging and statistics. This work is funded by an Army Research Office Young Investigator Award # W911NF-17-1-0346, a Burroughs

Wellcome Fund Career Award at the Scientific Interface, and The Johns Hopkins University Whiting School of Engineering start-up funds to CL.

## Additional information

### Funding

| Funder | Grant reference number | Author |
| --- | --- | --- |
| Army Research Office | W911NF-17-1-0346 | Chen Li |
| Burroughs Wellcome Fund | 1014584.01 | Chen Li |
| Johns Hopkins University | | Chen Li |

The funders had no role in study design, data collection and interpretation, or the decision to submit the work for publication.

### Author contributions

Ratan Othayoth, Conceptualization, Resources, Data curation, Software, Formal analysis, Validation, Investigation, Visualization, Methodology, Writing - original draft, Writing - review and editing; Chen Li, Conceptualization, Resources, Supervision, Funding acquisition, Validation, Methodology, Project administration, Writing - review and editing

### Author ORCIDs

Ratan Othayoth (iD) https://orcid.org/0000-0001-5431-9007
Chen Li (iD) https://orcid.org/0000-0001-7516-3646

### Decision letter and Author response

Decision letter https://doi.org/10.7554/eLife.60233.sa1
Author response https://doi.org/10.7554/eLife.60233.sa2

## Additional files

### Supplementary files

• Transparent reporting form

### Data availability

Data and code are made available online and can be accessed at: https://github.com/Terradynamics-Lab/self_righting (copy archived at https://archive.softwareheritage.org/swh:1:rev:e3369b9df138c75d0e490be0c48c53ded3e3a1d6).

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
