## [Decision Letter]

**Acceptance summary:**

This work studies the self-righting behavior in discoid cockroaches, with a robot model to test and evaluate the inspired strategy, which involves the movement of both wings and legs. It was found that the opening of the wings can change the potential energy landscape of the robot to lower the roll energy barrier so that the small kinematic fluctuation generated by the flailing of the legs can break this barrier with a higher probability. The authors also show that the phase coordination between the wings and the legs is critical to facilitate a successful righting, while the randomness in leg movements can help the robot to better explore the phase space and therefore find a proper leg-wing phase to right successfully. Thus, they have shown how noise and dynamics can interact to generate successful locomotion in a complicated landscape.

**Decision letter after peer review:**

Thank you for submitting your article "Appendage flailing facilitates strenuous ground self-righting" for consideration by *eLife*. Your article has been reviewed by three peer reviewers, including Gordon J. Berman as the Reviewing Editor and Reviewer #1, and the evaluation has been overseen by Christian Rutz as the Senior Editor. The following individual involved in the review of your submission has agreed to reveal their identity: Gabor Domokos (Reviewer #3).

The reviewers have discussed their reviews with one another, and the Reviewing Editor has drafted this decision to help you prepare a revised submission.

We would like to draw your attention to changes in our revision policy that we have made in response to COVID-19 (https://elifesciences.org/articles/57162).

Summary:

This work studies the self-righting behavior in discoid cockroaches, with a robot model to test and evaluate the inspired strategy, which involves co-actuation of both wings and leg. It was found that the opening of the wings can change the potential energy landscape of the robot as to lower the roll energy barrier so that the small kinematic energy fluctuation generated by the flailing of the legs can break this barrier with higher probability. In two separate papers attached to this submission, using the robot, the authors also show that the phase coordination between the wings and the leg is critical to facilitate a successful righting, while the randomness in leg movements can help the robot to better explore the phase space and therefore find a proper leg-wing phase to right successfully.

Essential revisions:

1) The authors assumed that the wing-leg coordination is completely open-loop, as the leg motion is either randomly generated or prescribed. It is unclear that the authors provide sufficient evidence to reject the hypothesis that the observed leg-wing motion is a result of certain reflexes with feedback control. From the results and the videos, it seems that the leg movements of the cockroach were neither periodic nor completely random, and therefore, it is likely that there could exist certain reflexes. One possibility is that the authors could quantify the leg kinematics and provide statistical analyses on the correlations between body, wing, and leg movements, thereby addressing the relative randomness within this behavior. Without such quantifications, the claim that the movement is "probabilistic" is not justified.

2) Although the reviewers liked the landscape metaphor for describing the stable and meta-stable points of the behavior, the authors need to include some language in the text that delineates their approach from more standard statistical mechanics dynamics amidst an energy landscape. For instance, the trajectories in Figure 5 do not behave like a thermal particle in a potential well. The trajectories, rather, lie along stereotyped lines in the space (even within the basins). This finding reflects the fact that there are other constraints that are not explicitly accounted for in the landscape (e.g., control system dynamics). While not problematic to the manuscript's overall message, it is important to explicitly distinguish between the landscape here and the landscape picture that most physicists will have in their heads.

3) As far as the referees understood, the potential energy function (Figure 4) is a one-parameter family of functions, with the parameter being the wing opening. At each wing opening angle, the authors seem to regard the convex hull H of the insect/robot, parametrize the hull H by two angles and the potential energy is the height of the center of mass for any given pair of angles. It is confusing that this parameterization appears to be non-unique, i.e. one position of the robot is mapped onto several points in the parameter plane. Clarification as to this point would be helpful.

4) Based on the plots Figure 4B and the description and the videos, it appears that the authors believe that for all wing opening angles, H always has just S=2 stable equilibrium points. If this is the case (i.e., S=2 for all wing openings), then this should be stated explicitly because this is far from trivial. If S=2, then shouldn't the two basins of attraction be defined by one periodic curve?

5) Some discussion of the scaling properties of the robot compared to the insect would be helpful. Given the fact that the robot is many times larger than a typical cockroach (~100x more massive), does a qualitatively similar energy landscape emerge for the parameters of an animal? Would we see the same metastable and stable states for realistic leg weights/lengths? For example, the robot has a largely exaggerated leg that is disproportionally heavier than those of the insects; actuating a large leg mass requires motor power that might be infeasible for most of the insects, and also could make the breaking of energy barrier more easy than that of insects (with the help of larger leg momentum and power from leg motors). Therefore, the successful strategy for the proposed robot may not be representative of those used by the cockroach. Also, the mass distribution and the dynamics of the robot and the cockroach seem quite different. Thus, the authors need to perform additional calculations to justify the claimed similarities between the robot and the insect.

---

## [Author Response]

Essential revisions:1) The authors assumed that the wing-leg coordination is completely open-loop, as the leg motion is either randomly generated or prescribed. It is unclear that the authors provide sufficient evidence to reject the hypothesis that the observed leg-wing motion is a result of certain reflexes with feedback control. From the results and the videos, it seems that the leg movements of the cockroach were neither periodic nor completely random, and therefore, it is likely that there could exist certain reflexes. One possibility is that the authors could quantify the leg kinematics and provide statistical analyses on the correlations between body, wing, and leg movements, thereby addressing the relative randomness within this behavior. Without such quantifications, the claim that the movement is "probabilistic" is not justified.

We thank the reviewer for the great suggestion and have added this analysis and discussion.

“In addition, wing opening and leg flailing did not show temporal correlation, indicating that leg flailing was more feedforward-driven rather than a feedback-controlled reflex coordinated with wing opening (Figure 3—Figure supplement 2). Moreover, large trial-to-trial variations in the number of attempts required to self-right showed that the animal’s self-righting was stochastic (Figure 3—Figure supplement 3).”

“Relationship between wing opening and leg flailing

We examined whether the animal’s leg flailing during self-righting was more feedforward-driven or more towards a feedback-controlled reflex coordinated with wing opening. […] This suggested that leg flailing had some rhythm, despite a large temporal variation and difference between the two hind legs (F. Delcomyn, 1987; Sherman et al., 1977; Zill, 1986).”

2) Although the reviewers liked the landscape metaphor for describing the stable and meta-stable points of the behavior, the authors need to include some language in the text that delineates their approach from more standard statistical mechanics dynamics amidst an energy landscape. For instance, the trajectories in Figure 5 do not behave like a thermal particle in a potential well. The trajectories, rather, lie along stereotyped lines in the space (even within the basins). This finding reflects the fact that there are other constraints that are not explicitly accounted for in the landscape (e.g., control system dynamics). While not problematic to the manuscript's overall message, it is important to explicitly distinguish between the landscape here and the landscape picture that most physicists will have in their heads.

To avoid confusion, we removed the analogy with noise-activated escape of a particle entirely. We explicitly state that our landscape is from gravitational potential energy.

“The landscape is the gravitational potential energy of the robot in its body pitch-roll space.”

“Model definition

The gravitational potential energy of the animal or robot is:

*E* = *mgz*_CoM_ (11)”

We added a discussion about the stereotyped trajectories being constrained by physical interaction.

“Stereotyped motion emerges from physical interaction constraint

Our landscape modeling demonstrated that the stereotyped body motion during strenuous leg-assisted, winged self-righting in both the animal and robot is strongly constrained by physical interaction of the body and appendages with the environment. […] Similarly, it will inform the design and control of self-righting robots (e.g., (Caporale et al., 2020; Kessens et al., 2012)).

3) As far as the referees understood, the potential energy function (Figure 4) is a one-parameter family of functions, with the parameter being the wing opening. At each wing opening angle, the authors seem to regard the convex hull H of the insect/robot, parametrize the hull H by two angles and the potential energy is the height of the center of mass for any given pair of angles. It is confusing that this parameterization appears to be non-unique, i.e. one position of the robot is mapped onto several points in the parameter plane. Clarification as to this point would be helpful.

The redundancy in parametrization arises from the Euler angle convention used for describing body pitch and roll and is inherent to any Euler angle-based description. For example, the same upright state of the robot can be described by both of following pairs of body pitch and roll (pitch, roll) = (0°, −180°) and (0°, 180°) (see Author response image 1). In addition, the rotation description using Euler angles are periodic as:

**Author response image 1. respfig1:** Potential energy landscape for an extended body pitch-roll space. Landscape shown for a wing opening angle of 51.2°. Black square shows the range of landscape used in the paper for analysis. Panels show the statically stable equilibrium configurations of the robot. Black arrows point to body pitch and roll corresponding to stable orientations.

*R* = *EulerToR*(roll, pitch, yaw) = *EulerToR*(roll ± 2*a*π, pitch ± 2*b*π, yaw ± 2*c*π)

where *R* is the rotation matrix, *EulerToR* is the function mapping roll, pitch, and yaw to the rotation matrix, and *a*, *b*, and *c* are integers. This periodicity can be seen by calculating the potential energy landscape for an extended range (see Author response image 1).

However, this does not affect our modeling and conclusions, because the system state only reaches a redundant state towards the end of self-righting when it is near-upright (see Footnote 6).

4) Based on the plots Figure 4B and the description and the videos, it appears that the authors believe that for all wing opening angles, H always has just S=2 stable equilibrium points. If this is the case (i.e., S=2 for all wing openings), then this should be stated explicitly because this is far from trivial. If S=2, then shouldn't the two basins of attraction be defined by one periodic curve?

In fact, the robot can have more than two static equilibria. For example, at a wing opening angle of 51.2°, in addition to the upside-down, metastable, and upright states shown in Figure 5, there are also four other system states where the robot is in static equilibrium (see Author response image 1).

In addition, the number of stable equilibria also varies with wing opening angle. For example, except for the metastable and the three upright equilibria described in the paper (and shown above), other equilibria (shown in Author response image 1) disappear when wings are closed. The landscape visualization in MATLAB supplied as a part of data (TerradynamicsLab/self_righting (github.com)) may be used to examine these robot configurations. However, these are shallow basins within a deeper basin and do not qualitatively affect the self-righting transitions and hence we do not consider them in the analysis.

Attractive basins of the landscape (which does not yet describe dynamics, see Lines 279-288) is not same as basins of attraction of a dynamical system. Thus, we do not expect the basin to be defined by a periodic curve.

5) Some discussion of the scaling properties of the robot compared to the insect would be helpful. Given the fact that the robot is many times larger than a typical cockroach (~100x more massive), does a qualitatively similar energy landscape emerge for the parameters of an animal? Would we see the same metastable and stable states for realistic leg weights/lengths? For example, the robot has a largely exaggerated leg that is disproportionally heavier than those of the insects; actuating a large leg mass requires motor power that might be infeasible for most of the insects, and also could make the breaking of energy barrier more easy than that of insects (with the help of larger leg momentum and power from leg motors). Therefore, the successful strategy for the proposed robot may not be representative of those used by the cockroach. Also, the mass distribution and the dynamics of the robot and the cockroach seem quite different. Thus, the authors need to perform additional calculations to justify the claimed similarities between the robot and the insect.

We thank the reviewer for the great suggestion and have added this analysis and discussion.

“Considering that body rolling is induced by centrifugal force from leg flailing, we compared the ratio of leg centrifugal force to leg gravitational force between the animal and robot and verified they are dynamically similar (see Materials and methods for detail). […] Thus, the physical principles discovered for the robot are applicable to the animal.”

“Similar to the animal, the robot’s self-righting was stochastic, with large trial-to-trial variation in the number of attempts required to self-right and body pitching and rolling motions (Figure 6, Figure 6—Figure supplement 1).”

“Robot and animal have similar evolving potential energy landscapes

For both the animal and robot, the potential energy landscape over body pitch-roll space were similar in shape, and both changed in a similar fashion as the wings opened (Figure 5, Figure 5—Figure supplement 1). […] To self-right via either the pitch (Figures 5Aiii-iv, 1Aiii-iv) or roll (Figures 5Aiii’-iv’, 1Aiii’-iv’) mode, the system state must escape from the metastable basin to reach either the pitch or a roll upright basin (e.g., Figure 5Biii, blue and red curves).”

“Similarity to animal

To examine whether the robotic physical model was similar to the animal and reasonably approximated its self-righting motion, we examined how well they were geometrically similar and their leg flailing motions were dynamically similar. […] This dynamic similarity demonstrated that the robot provided a good physical model for studying the animal’s self-righting.”